# Embryologic Origin of the Primary Tumor and RAS Status Predict Survival after Resection of Colorectal Liver Metastases

**DOI:** 10.3390/medicina58081100

**Published:** 2022-08-14

**Authors:** Sorin Tiberiu Alexandrescu, Ioana Mihaela Dinu, Andrei Sebastian Diaconescu, Alexandru Micu, Evelina Pasare, Cristiana Durdu, Bogdan Mihail Dorobantu, Irinel Popescu

**Affiliations:** 1Department of General Surgery, Fundeni Clinical Institute, 022328 Bucharest, Romania; 2Faculty of Medicine, Carol Davila University of Medicine and Pharmacy, 050474 Bucharest, Romania; 3Department of Oncology, Fundeni Clinical Institute, 022328 Bucharest, Romania; 4Filantropia Clinical Hospital, 011171 Bucharest, Romania

**Keywords:** colorectal liver metastases, RAS mutational status, primary tumor sidedness, embryologic origin, liver resection, survival after recurrence

## Abstract

*Background and objectives*. In colorectal cancers, the embryologic origin of the primary tumor determines important molecular dissimilarities between right-sided (RS) and left-sided (LS) carcinomas. Although important prognostic differences have been revealed between RS- and LS-patients with resected colorectal liver metastases (CLMs), it is still unclear if this observation depends on the RAS mutational status. To refine the impact of primary tumor location (PTL) on the long-term outcomes of patients with resected CLMs, the rates of overall survival (OS), relapse-free survival (RFS) and survival after recurrence (SAR) were compared between RS- vs. LS-patients, according to their RAS status. *Material and Methods*. All patients with known RAS status, operated until December 2019, were selected from a prospectively maintained database, including all patients who underwent hepatectomy for histologically-proven CLMs. A log-rank test was used to compare survival rates between the RS- vs. LS-group, in RAS-mut and RAS-wt patients, respectively. A multivariate analysis was performed to assess if PTL was independently associated with OS, RFS or SAR. *Results*. In 53 patients with RAS-mut CLMs, the OS, RFS and SAR rates were not significantly different (*p* = 0.753, 0.945 and 0.973, respectively) between the RS and LS group. In 89 patients with RAS-wt CLMs, the OS and SAR rates were significantly higher (*p* = 0.007 and 0.001, respectively) in the LS group vs. RS group, while RFS rates were similar (*p* = 0.438). The multivariate analysis performed in RAS-wt patients revealed that RS primary (*p* = 0.009), extrahepatic metastases (*p* = 0.001), N-positive (*p* = 0.014), age higher than 65 (*p* = 0.002) and preoperative chemotherapy (*p* = 0.004) were independently associated with worse OS, while RS location (*p* < 0.001) and N-positive (*p* = 0.007) were independent prognostic factors for poor SAR. *Conclusions*. After resection of CLMs, PTL had no impact on long-term outcomes in RAS-mut patients, while in RAS-wt patients, the RS primary was independently associated with worse OS and SAR.

## 1. Introduction

Colorectal cancer is the most frequent digestive malignancy worldwide, accounting for almost 10% of cancer-related deaths [1]. The main cause of death in patients with colorectal cancer is metastatic disease. Although resection of primary tumor and metastases enables 5-year overall survival rates higher than 25% in most series reported so far, prognosis depends on clinical, pathologic, molecular and genetic factors. The embryologic origin of the primary colorectal cancer determines important molecular and pathologic dissimilarities between right-sided (RS) and left-sided (LS) carcinomas [2]. These biologic differences seem to have prognostic implications for tumors derived from midgut (RS tumors—located between caecum and splenic flexure) and those originating in the hindgut (LS tumors—including descending and sigmoid colon, as well as rectum). During the last decade, most papers suggested that patients with unresectable metastatic RS colon carcinomas have lower overall survival (OS) rates compared to those with LS primaries [2,3,4]. In patients with resected colorectal liver metastases (CLMs), few reports failed to identify a significant impact of tumor sidedness on long-term outcomes [5,6,7]. Other studies, although they reported better OS rates after resection of CLMs in LS patients, revealed that LS tumors have similar [8,9] or even worse recurrence-free survival (RFS) rates compared to RS primaries [10]. In this fragmented landscape, some authors searched for a potential relationship between primary tumor location (PTL) and tumor mutational status. Although previous reports suggested that CLMs from RS colon cancer are associated with worse survival independent of KRAS mutational status [11,12,13], more recent studies suggested that a better prognostic stratification of patients with resected CLMs could be achieved by combining sidedness of the primary tumor and RAS mutational status [6,14,15]. RAS status is a well-known predictive factor for the response to anti-EGFR agents [16], but its prognostic value in patients with resected CLMs is still debatable. While some recent studies suggested that LS location is associated with better OS only in RAS-wt patients [14,17], the data about the impact of PTL on RFS rates according to the RAS status are lacking. Furthermore, the impact of primary tumor sidedness on survival after recurrence (SAR) in patients with resected CLMs has never been evaluated according to the RAS mutational status. To refine the impact of PTL on the long-term outcomes achieved by the resection of CLMs, we compared the rates of OS, RFS and SAR between LS and RS patients, according to their RAS status.

## 2. Materials and Methods

Since 2002, in our center, all patients who underwent curative-intent liver resection for suspected CLMs were prospectively enrolled in a database. The patients whose pathologic examination did not confirm the diagnosis of CLMs were excluded from the database. Preoperatively, the diagnosis of CLMs was established based on a contrast-enhanced CT scan, followed by a liver MRI when CT scan imaging was doubtful.

### 2.1. Selection of the Patients

All patients with a known RAS status, operated until December 2019, were selected from a prospectively maintained database, including all patients who underwent hepatectomy for histologically-proven CLMs in our center. Patients who died during the first 30 days post-surgery were excluded, because their death was most likely not secondary to cancer progression. Incomplete resection (R1/R2) and incomplete follow-up data were also exclusion criteria. Because the colon between the cecum and splenic flexure derives from midgut, patients with carcinomas with such a location were included in the right-sided group (RS group). Patients with primary colon tumors located distal to the splenic flexure and those with rectal carcinomas (which derive from hindgut) were included in the left-sided group (LS group). Patients with carcinomas located at the level of splenic flexure were excluded from the analysis. Similarly, patients with synchronous colorectal carcinomas located both on the right and left colon were excluded from analysis.

### 2.2. Molecular Diagnosis

Only mutations in exon 2 of the KRAS gene (at codons 12/13) were evaluated until 2013. After that, the tumors were subjected to full KRAS (exons 2, 3 and 4) and NRAS (exons 2, 3 and 4) analysis. Between 2006 and 2014, mutation detection was performed by High Resolution Melt Analysis (HRMA) and secondary confirmation by sequencing (ABI Prism 3130 sequencer). Since 2015, a targeted resequencing assay (Ion AmpliSeq© Panel, Thermo Fisher Scientific, Waltham, MA, USA) was used for mutation detection in exons 2, 3 and 4 of the KRAS and NRAS genes. Sequencing was carried out using the Next Generation Sequencing platform Ion proton (Thermo Fisher Scientific). Because treatment with monoclonal antibodies (either anti-EGFR or anti-VEGF agents) is recommended only in patients with unresectable metastatic colorectal cancer, in most patients RAS status was evaluated when they developed recurrence after resection of CLMs. Only in a small number of patients was RAS status evaluated immediately after liver resection (irrespective of recurrence development).

### 2.3. Treatment Allocation

Decisions regarding the timing of treatment were discussed in a multidisciplinary team. Patients with poor prognostic factors, such as extrahepatic disease, multiple, bilobar CLMs and initially unresectable or borderline resectable metastases, typically underwent neoadjuvant therapy. In contrast, patients with favorable prognostic factors were typically sent for up-front surgery. CLMs were considered resectable when complete clearance of the liver was anticipated before operation and the volume of the remnant liver parenchyma exceeded 30% of the total liver volume. In patients with concomitant extrahepatic metastases, surgery was recommended when complete resection of hepatic and extrahepatic metastases had been anticipated. When complete resection of metastatic disease (either hepatic or extrahepatic) could not be achieved by one or staged procedures, the operation was considered an incomplete resection and these patients were excluded from the current analysis.

Postoperative (adjuvant) chemotherapy was recommended to all the patients. Targeted therapies have never been used in the adjuvant setting for patients with completely resected CLMs, according to the international guidelines.

Only the patients with unresectable metastases from colorectal origin could benefit from the treatment with monoclonal antibodies. Thus, patients included in this study (who underwent complete resection of metastatic disease) received treatment with monoclonal antibodies associated to chemotherapy, according to the current guidelines, only after the disease’s recurrence.

Although the patients enrolled in this study were treated during a long period of time (2006–2019), the oncologic therapy that was delivered to these patients was the same during this interval. Thus, chemotherapy consisted of a combination of 5-fluorouracil or capecitabine with oxaliplatin or irinotecan. During the entire period, targeted therapies consisted of anti-VEGF agent (bevacizumab) or anti-EGFR monoclonal antibodies (cetuximab or panitumumab). An assessment of the RAS status was specially performed to evaluate the possibility to give an anti-EGFR agent to the patient, because it was known since 2006 that tumors that have a RAS mutation do not respond to the anti-EGFR therapy. In such patients, the only targeted therapy available consisted of an anti-VEGF agent (bevacizumab).

### 2.4. Long-Term Outcomes

OS was calculated as the interval between liver resection and the date of patient’s death or the last follow-up (performed by personal contact with the patient, the patient’s family or the attending oncologist). RFS was calculated as the time period between hepatectomy and the date of malignancy recurrence or the last follow-up, if the patient was free of disease at that moment. SAR represents the interval between the recurrence of disease (after hepatectomy) and the death of the patient or the last follow-up (if the patient was alive at that moment). The patients who did not develop recurrent disease until the last follow-up were not included in the SAR analysis.

### 2.5. Prognostic Factors

To assess prognostic factors associated with long-term outcomes, the following parameters have been evaluated: age, sex, location of the primary tumor, presence of extrahepatic metastases, pathology data of colorectal tumor and liver metastases (pT, pN, maximum diameter, number and distribution of CLMs), tumor burden score (TBS), the association of ablative therapy concomitant with hepatectomy, the use of preoperative and postoperative chemotherapy as well as the presence and grade of postoperative complications. For patients with recurrent disease, the time interval between initial resection of metastases and the occurrence of recurrence was recorded. Furthermore, we recorded if the recurrent disease was resected or not. TBS was calculated according to the formula: TBS^2^ = (maximum tumor diameter)^2^ + (number of tumors)^2^, as previously published [18]. Postoperative complications were graded according to the Clavien–Dindo classification [19]. Minor complications were defined as Clavien–Dindo grades I or II, while major morbidity included complications graded III or higher according to the Clavien–Dindo classification.

### 2.6. Statistical Analysis

Categorical data are presented as absolute numbers and percentages. The association between categorical variables was analyzed with the Fischer exact test. Continuous data are presented as mean +/− standard deviation (SD) or as median and interquartile range [IQR25%-IQR75%], according to the tests used to evaluate the normality of distribution. Normality distribution was assessed by Shapiro–Wilk test and further comparison was performed with a t-test or Mann Whitney U test, accordingly. Survival rates were estimated with the Kaplan–Meier method and were compared between different groups by log-rank test. In a univariate analysis, the impacts of the previously mentioned parameters on OS, RFS and SAR were evaluated. The parameters that were associated with a *p*-value less than 0.10 at the univariate analysis were included in the multivariate analysis. A multivariate Cox proportional hazards regression analysis with a backward stepwise selection process was used to identify the independent prognostic factors associated with OS, RFS and SAR. A hazard ratio (HR) was reported with 95% confidence interval (CI). A *p* value lower than 0.05 was considered significant. The statistical analysis was performed using IBM SPSS software, version 23 (SPSS Inc, Chicago, IL, USA).

The study protocol, number 6571/1.02.2022, was approved by the local Institutional Review Board.

## 3. Results

There were 142 patients fulfilling the inclusion criteria and operated on between 2006 and 2019. Out of these, 53 had RAS-mut CLMs, while 89 had RAS-wt metastases.

### 3.1. RAS-Mut

There were 53 patients with resected RAS-wt CLMs fulfilling the inclusion criteria, with a mean age of 59.11 (+/−9.541) years old (*p* = 0.462, Shapiro–Wilk test). Out of these, 36 (67.9%) were male and 13 (24.5%) had RS primary tumors. In total, 7 patients (13.2%) had concomitant extrahepatic metastases (4—lymph nodes metastases, 2—limited peritoneal metastases and 1—ovarian metastases) that were completely resected concomitant with liver resection. Postoperative complications were recorded in 27 patients (50.9%), with 9 of them (17%) developing major morbidity. The comparative characteristics of patients with resected RAS-mut CLMs according to the primary sidedness are summarized in Table 1.

#### Long-Term Outcomes

For the entire group, the median OS was 31 months, with 1-, 3- and 5-year OS rates of 92.4%, 48.1% and 17.8%, respectively. The 1-, 3- and 5-year OS rates were not significantly different (*p* = 0.753) between patients with LS primary tumors (94.9%, 48.8% and 15.8%, respectively) and those with RS colorectal tumors (84.6%, 46.2% and 23.1%, respectively) (Figure 1a).

After a median follow-up of 31 months, 48 patients developed recurrence: hepatic only—23 patients; hepatic and extrahepatic—10 patients; lung—7 patients; peritoneal—2 patients; lymph nodes—2 patients; local recurrence—2 patients; ovarian—1 patient; and bone—1 patient. For the entire group, the median RFS was 10 months, with 1- and 3-years RFS rates of 33.6%, and 3.6%, respectively. The RFS rates were not statistically significant different between the LS group and RS group (33.6% and 5.9% vs. 34.2% and 0% at 1- and 3-years, respectively, *p* = 0.945) (Figure 1b).

For all the patients who developed recurrence after the initial resection of CLMs, the 1-, 3- and 5-year SAR rates were 89.4%, 20.4% and 10.3%, respectively (median 24 months). The rate of SAR was similar in the LS group and RS group (94.3%, 18.6% and 11.1% vs. 75%, 25% and 0% at 1-, 3- and 5-years, respectively, *p* = 0.973) (Figure 1c).

### 3.2. RAS-Wt

There were 89 patients with resected RAS-wt CLMs fulfilling the inclusion criteria, with a median age of 57.53 (+/−9.169) years old (*p* = 0.075, Shapiro–Wilk test). Out of these, 50 were male (56.2%) and 15 (16.9%) had RS primary tumors. In total, 10patients (11.2%) had concomitant extrahepatic metastases (5—limited peritoneal metastases, 3—hepatic pedicle lymph nodes metastases and 2—lung metastases) that were completely resected concomitant with CLMs (9 patients) or before hepatectomy (1 patient with lung metastases that were resected 3 months previous to liver resection). Postoperative complications were recorded in 39 patients (43.8%), with 12 of them developing major morbidity. The baseline characteristics of the patients are presented in Table 1. There were not significant differences between the LS group and RS group regarding baseline characteristics, the interval between resection of CLMs and disease recurrence or the resectability of recurrence (Table 1).

#### 3.2.1. Long-Term Outcomes

For the entire group, the median OS was 45 months, with 1-, 3- and 5-year OS rates of 95.5%, 58.2% and 26.6%, respectively. In patients with LS primary tumors, the 1-, 3- and 5-year OS rates (97.3%, 62.5% and 28.4%, respectively) were significantly higher (*p* = 0.007) than those achieved by liver resection in the RS group (86.7%, 36.1% and 10.8%, respectively) (Figure 2a).

After a median follow-up of 39 months, 78 patients developed recurrences: hepatic only—42 patients; hepatic and extrahepatic—15 patients; pulmonary—8 patients; peritoneal—4 patients; lymph nodes—4 patients; pelvic recurrence—3 patients; ovarian—1 patient; and bone—1 patient. The recurrence rate was not significantly different between the RS group (13/15) and LS group (65/74) (*p* = 0.899). For the entire group, the median RFS was 11 months, with 1- and 3-years RFS rates of 38.6% and 12.7%, respectively. The RFS rates were not statistically significant different between the LS group and RS group (40.2% and 8.1% vs. 30% and 15% at 1- and 3-years, respectively, *p* = 0.438) (Figure 2b).

Recurrent disease developed during the first year after initial resection of CLMs in 52 patients (66.7%) and after more than one year in 26 patients (33.3%). The recurrence was resected in 27 patients (34.6%): hepatic re-resection—17 patients; lung resection—6 patients; hepatic and extrahepatic resection—2 patients; oophorectomy—1 patient; and hepatic pedicle lymph nodes dissection—1 patient. Although the resectability rate of recurrence was higher in the LS group (25/65—38.4%) than in the RS group (2/13—15.3%), the difference was not statistically significant (*p* = 0.199).

For all the patients who developed recurrence after initial resection of CLMs, the 1-, 3- and 5-year SAR rates were 87.1%, 38.1% and 10%, respectively (median 33 months). The rates of SAR were significantly higher in the LS group vs. RS group (87.5%, 45.5% and 12% vs. 68.4%, 8.5% and 0% at 1-, 3- and 5-years, respectively, *p* < 0.001) (Figure 2c).

#### 3.2.2. Univariate Analysis

Factors associated with a significantly worse OS in the univariate analysis were the RS location of the primary tumor (*p* = 0.007), extrahepatic metastases (*p* = 0.014) and metastatic lymph nodes around the primary tumor (N+) (*p* = 0.004). Age higher than 65 (*p* = 0.095) and the use of preoperative chemotherapy (*p* = 0.084) were marginally associated with poor OS in the univariate analysis (Table 2).

The presence of postoperative complications (*p* = 0.024), extrahepatic disease (*p* = 0.003) and multiple CLMs (*p* = 0.026) were associated with significantly lower RFS rates in the univariate analysis (Table 3).

In the univariate analysis, the factors significantly associated with lower SAR rates were RS tumors (*p* < 0.001), N-positive primary tumor (*p* = 0.011), appearance of the recurrence during the first 12 months after resection of CLMs (*p* = 0.048) and resection of recurrence (*p* = 0.007) (Table 4).

#### 3.2.3. Multivariate Analysis

To identify independent prognostic factors for poor long-term outcomes, characteristics that were associated with a *p* value < 0.01 in the univariate analysis were included in the multivariate analysis. Factors that were independently associated with poor OS were RS location of the primary tumor (*p* = 0.009), extrahepatic metastases (*p* = 0.001), N-positive primary tumor (*p* = 0.014), age higher than 65 years old (*p* = 0.002) and the use of preoperative chemotherapy (*p* = 0.004) (Table 2). For RFS, the factors independently associated with poor prognosis were postoperative complications (*p* = 0.024) and extrahepatic metastases (*p* = 0.015) (Table 3). RS tumors (*p* < 0.001) and N-positive status of the primary tumor (*p* = 0.007) were the only independent prognostic factors for poor SAR (Table 4).

## 4. Discussion

The data presented here argue that the prognostic stratification of patients with resected CLMs can be refined by using a combination of PTL and RAS status. Although some studies reported that liver resection achieved better OS rates in patients with LS primary tumors compared to those with RS colorectal carcinomas [8,9,10,11,12,13,20], other studies failed to identify a significant association between PTL and OS [5,6,7]. A meta-analysis revealed that although the RS location of the primary tumor was overall associated with poor OS, almost half of the included studies did not show significant associations between RS tumors and worse OS [21]. An explanation for these conflicting results has been suggested by a single center study published in 2020 [15], supported by a multi-center study published in 2021 [14] and strengthened by a meta-analysis published in 2022 [17], which revealed that the prognostic impact of PTL depends on the RAS status. Thus, the LS location of the primary tumor was associated with significantly better OS rates after hepatectomy for CLMs only in KRAS-wt tumors, while in patients with KRAS-mut CLMs there was not any significant difference in OS according to the PTL [14,15,17]. A similar result is reported in the present series, with PTL being independently associated with OS only in patients with resected RAS-wt CLMs. In contrast, in RAS-mut CLMs, we did not find a significant difference in OS rates in LS vs. RS patients. That observation can explain how different proportions of RAS-mut/RAS-wt patients included in previous studies which evaluated the impact of PTL regardless of the RAS status might induce a bias in the analysis of the impact of PTL on OS in these cohorts, resulting in the heterogeneous results that have been reported [22]. Thus, the higher proportion of RAS-wt CLMs observed in LS patients [6,7] might tip the scale toward higher OS rates in the LS group in those studies that evaluated the impact of PTL, irrespective of the RAS status.

Regarding the impact of PTL on RFS, previous studies generated even more conflicting results [8,9,10,23]. For example, 19 out of 25 studies included in a meta-analysis published in 2019 did not find a significant association between PTL and RFS [21]. Although the results of this meta-analysis revealed a marginally significant prognostic role of PTL regarding RFS, the authors concluded that the prognostic value of PTL on RFS should be regarded with caution, as long as its effect in predicting RFS was limited [21]. However, in the above-mentioned meta-analysis, the impact of PTL was not evaluated in correlation with RAS status. In light of the more recent evidence that the PTL impacts OS after resection of CLMs only in RAS-wt tumors, one may hypothesize that the results of this meta-analysis were altered by the inclusion of patients, irrespective of their RAS status. The present study refined the prognostic impact of PTL on RFS according to the RAS status, revealing that PTL has not a significant influence on RFS neither in RAS-mut CLMs, nor in RAS-wt ones. In the current study, the only factors which were independently associated with poor RFS in RAS-wt patients were the presence of extrahepatic metastases (*p* = 0.015) and development of complications after hepatectomy (*p* = 0.024). These two variables were also reported as independent prognostic factors for poor RFS in many other studies [5,24,25].

Although two previous studies revealed that patients with RS tumors had worse SAR, the authors did not investigate whether this observation is independent or not of RAS mutational status [8,10]. To address this question, the current study also investigated the impact of the embryologic origin of colorectal cancer on SAR in patients with resected CLMs, according to the RAS status. Although in RAS-mut patients the sidedness of the primary tumor had no impact on SAR, in patients with RAS-wt CLMs, the PTL had been independently associated with SAR (*p* < 0.001). One could hypothesize that better SAR rates in the LS group may be due to a higher resectability rate of the recurrence in these patients. This hypothesis cannot be supported by our results, as long as the resectability rates were not significantly different between the two groups. Moreover, although in the univariate analysis the resection of recurrent disease has been significantly associated (*p* = 0.007) with better SAR, it was not independently associated with SAR in the multivariate analysis. These observations rather suggest the reduced efficacy of current oncologic therapies in patients with RS primary colon cancer who develop recurrence after resection of RAS-wt CLMs, compared to those with LS tumors. A similar finding has been reported by a recent study, which revealed that in RAS-wt patients, primary tumor sidedness was strongly associated with OS, irrespective of the type of biological agent that was used (EGFR-inhibitor or bevacizumab) [26]. The above-mentioned study revealed that despite receipt of an EGFR-inhibitor or bevacizumab, sidedness plays the most important role in OS of patients with unresected CLMs [26]. These results are in line with those of a recent meta-analysis of 12 randomized trials, which found that in medically treated patients with unresectable metastatic colorectal cancer, the prognostic value of PTL was restricted to the KRAS wild-type population [27]. To the best of our knowledge, the current study is the first one that investigated the impact of tumor sidedness on SAR in patients with resected CLMs according to the RAS mutational status. Our results disclose for the first time that the better OS observed in the LS group after resection of RAS-wt CLMs is mainly attributable to a significantly higher SAR rate in this group of patients.

Since the RFS rates after hepatectomy are similar irrespective of PTL and RAS status, liver resection should not be discouraged in patients with RS primary tumors. The lower OS and SAR rates achieved by onco-surgical approaches in patients with RS primary tumors and RAS-wt CLMs rather emphasize the need for more efficient oncologic therapies in these patients.

Several observations could explain the worse long-term outcomes of patients with RS colon cancer and RAS-wt CLMs. Because RAS- and BRAF-mutations are typically mutually exclusive in patients with CLMs, it could be estimated that up to 10% of patients with RAS-wt CLMs from this series harbor a BRAF mutation [28,29]. In patients with metastatic colorectal cancer, BRAF-mut portends a significantly worse prognosis [11,28,29,30], and is associated with both a lack of response to anti-EGFR therapy [31] and a decreased resectability rate of recurrence after the initial resection of CLMs [32]. As RS primary tumors are more frequently associated with BRAF-mutations than the LS colorectal cancers [11,28], that can tip the balance of OS and SAR in favor of the LS group. Furthermore, RS carcinomas are more likely associated with microsatellite instability (MSI) than LS colorectal cancers [11,33]. Regarding the impact of MSI on the long-term outcomes of patients who underwent surgery for CLMs, a recent study including patients treated with resection and/or ablation for CLMs revealed that OS was significantly lower in the MSI group (*p* < 0.001), while local or distant progression-free survival rates were not significantly different between the MSI and MSS groups [34]. The lack of evaluation for other molecular alterations (e.g., BRAF status, MSI status, TP53, etc.) in the current series represent a limitation of this study. Thus, this study cannot draw a definitive conclusion on the mechanisms that determine a different prognosis in patients with resected CLMs according to the PTL, but could offer a basis for including patients with resected CLMs in distinct prognostic groups in future studies aiming to assess the molecular basis that determines dissimilar survival outcomes in LS vs. RS patients.

Another limitation of this study is its retrospective nature. For example, patients operated until 2013 were not tested for NRAS and KRAS exon 3,4 mutations, and consequently, some of them could be misclassified as RAS-wt. Because only 19 RAS -wt patients included in this study were operated on between 2006 and 2012 and the rate of additional RAS mutations is less than 15% in KRAS exon 2-wt patients [35], we estimate that up to 5 patients from this cohort might be misclassified as RAS-wt. It is unlikely that such a small proportion of misclassified patients (3.5%) could lead to an important bias in survival rates. Another limitation that could be perceived for this study is that the RAS mutational status was assessed, in most patients, at the time of recurrence development after initial hepatectomy. This is due to the fact that the use of monoclonal antibodies is not recommended in the adjuvant setting after the complete resection of CLMs, being indicated only in the palliative therapy of patients with CLMs. Thus, a lot of the patients operated in our center who did not develop recurrence after hepatectomy were not included in this study, as their RAS mutational status was not evaluated. That may explain the lower OS and RFS rates reported in this cohort, but had no influence on SAR rates. One could consider that another shortcoming of this study is the inclusion of the left colon cancers and rectal adenocarcinomas in the same group. The reason was their common embryologic origin and their similar outcomes compared to right-sided colon carcinomas. Although a recent study suggested that OS rates achieved by metastasectomy for CLMs were similar in patients with rectal cancers and right-sided primaries [15], most studies dealing with this subject reported similar long-term outcomes achieved by liver resection for CLMs in patients with left-sided colon cancers and rectal carcinomas [3,34,36]. Furthermore, a meta-analysis published in 2022 revealed that the variable effect of KRAS status on PTL persisted, regardless of whether the patients with rectal tumors were included or not in the LS group [17].

## 5. Conclusions

The effect of the embryologic origin of colorectal cancers on long-term outcomes after the resection of CLMs depends on the RAS mutational status. In RAS-mut CLMs, the primary tumor location has no impact on long-term outcomes. In RAS-wt patients, the RS location of the primary is independently associated with poorer OS and SAR, but not with RFS. In patients with resected CLMs, PTL enables prognostic stratification only for RAS-wt tumors.

## Figures and Tables

**Figure 1 medicina-58-01100-f001:**
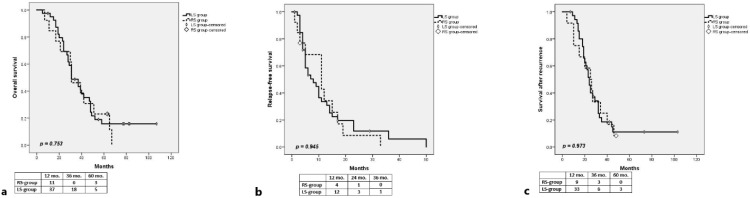
Comparative long-term outcomes between LS group and RS group in patients with RAS-mut CLMs; (**a**) overall survival; (**b**) relapse-free survival; (**c**) survival after recurrence.

**Figure 2 medicina-58-01100-f002:**
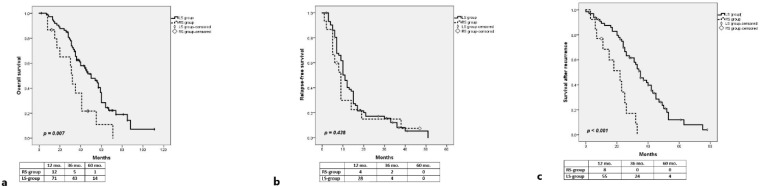
Comparative long-term outcomes between LS group and RS group in patients with RAS-wt CLMs; (**a**) overall survival; (**b**) relapse-free survival; (**c**) survival after recurrence.

**Table 1 medicina-58-01100-t001:** Comparative characteristics of RAS-wt patients, according to the primary tumor location.

Clinico-Pathologic Characteristics	Right-Sided Group	Left-Sided Group	*p* Value (*t* test/Fischer’s Exact Test)
	*N* (%)	*N* (%)	
Age (mean +/− SD)	58.47 (+/−10.81)	57.34 (+/−8.87)	0.6667
Sex			0.5731
Male	8 (53.3%)	32 (43.2%)	
Female	7 (46.7%)	42 (56.8%)	
Postoperative complications			1
No	8 (53.3%)	42 (56.8%)	
Yes	7 (46.7%)	32 (43.2%)	
Major complications			1
No	13 (86.7%)	64 (86.5%)	
Yes	2 (13.3%)	10 (13.5%)	
Extrahepatic metastases			0.674
No	13 (86.7%)	66 (89.2%)	
Yes	2 (13.3%)	8 (10.8%)	
Extension of hepatectomy			1
Minor	12 (80%)	56 (78.4%)	
Major	3 (20%)	16 (21.6%)	
Associated ablation			0.3871
No	12 (80%)	66 (89.2%)	
Yes	3 (20%)	8 (10.8%)	
Maximum size of CLMs			0.219
≤5 cm	9 (60%)	56 (75.7%)	
>5 cm	6 (40%)	18 (24.3%)	
Maximum size of CLMs			0.2704
≤3 cm	5 (33.3%)	37 (50%)	
>3 cm	10 (66.7%)	37 (50%)	
Number of CLMS			0.7759
Single	7 (46.7%)	30 (40.5%)	
Multiple	8 (53.3%)	44 (59.5%)	
Number of CLMS			1
<4	11 (73.3%)	51 (68.9%)	
≥4	4 (26.7%)	23 (31.3%)	
TBS			1
≤4.47	7 (46.7%)	36 (48.6%)	
>4.47	8 (53.3%)	38 (51.4%)	
CLMs’ distribution			0.5832
Unilobar	9 (60%)	38 (51.4%)	
Bilobar	6 (40%)	36 (48.6%)	
T-status			0.6786
T1–T3	12 (80%)	55 (74.3%)	
T4	1 (6.7%)	11 (14.9%)	
NA	2 (13.3%)	8 (10.8%)	
N-status			0.744
N−	3 (20%)	21 (28.4%)	
N+	10 (66.7%)	45 (60.8%)	
NA	2 (13.3%)	8 (10.8%)	
Synchronous vs. Metachronous			0.5585
Synchronous	11 (73.3%)	46 (62.2%)	
Metachronous	4 (26.7%)	28 (37.8%)	
Initially resectable CLMs			0.4933
Yes	11 (73.3%)	60 (81.1%)	
No	4 (26.7%)	14 (18.9%)	
Preoperative chemotherapy			0.3958
No	10 (66.7%)	38 (51.4%)	
Yes	5 (33.3%)	36 (48.6%)	
Adjuvant chemotherapy			1
Yes	14 (93.3%)	68 (91.9%)	
No	1 (6.7%)	4 (5.4%)	
NA		2 (2.7%)	
Resection of recurrence *			0.1998
No	11 (84.6%)	40 (61.5%)	
Yes	2 (15.4%)	25 (38.5%)	
Time to recurrence *			0.526
≤12 months	10 (76.9%)	42 (64.6%)	
>12 months	3 (23.1%)	23 (35.4%	

* Malignancy recurred in 78 patients (11 patients did not develop recurrence during follow-up).

**Table 2 medicina-58-01100-t002:** Univariate and multivariate analysis for OS in RAS-wt patients.

Clinico-Pathologic Characteristics	*p* Value (Univariate, Log-Rank)	HR (Multivariate, Cox Regression)	95% CI (Multivariate, Cox Regression)	*p* Value (Multivariate, Cox Regression)
Age	0.095			** *0.002* **
≤65 y-o		1		
>65 y-o		0.292	0.133–0.640	
Sex	0.743			
Female				
Male				
Right vs. Left	*0.007*			** *0.009* **
Right-sided		0.398	0.199–0.794	
Left-sided		1		
Postoperative complications	0.672			
No				
Yes				
Major complications	0.305			
No				
Yes				
Extrahepatic metastases	*0.014*			** *0.001* **
No		1		
Yes		0.27	0.125–0.684	
Extension of hepatectomy	0.109			
Minor				
Major				
Associated ablation	0.463			
No				
Yes				
Maximum size of CLMs	0.394			
≤5 cm				
>5 cm				
Maximum size of CLMs	0.88			
≤3 cm				
>3 cm				
Number of CLMS	0.324			
Single				
Multiple				
Number of CLMS	0.628			
<4				
≥4				
TBS	0.105			
≤4.47				
>4.47				
CLMs’ distribution	0.765			
Unilobar				
Bilobar				
T-status	0.274			
T1-T3				
T4				
N-status	*0.004*			** *0.014* **
N−		1		
N+		0.426	0.216–0.841	
Synchronous vs. Metachronous	0.366			
Synchronous				
Metachronous				
Initially resectable CLMs	0.396			
Yes				
No				
Preoperative chemotherapy	0.084			** *0.004* **
No		1		
Yes		0.44	0.251–0.774	
Adjuvant chemotherapy	0.939			
Yes				
No				

Italic-bold: for all values lower than 0.05 (in this column).

**Table 3 medicina-58-01100-t003:** Univariate and multivariate analysis for RFS in RAS-wt patients.

Clinico-Pathologic Characteristics	*p* Value (Univariate, Log-Rank)	HR (Multivariate, Cox Regression)	95% CI (Multivariate, Cox Regression)	*p* Value (Multivariate, Cox Regression)
Age	0.937			
≤65 y-o				
>65 y-o				
Sex	0.902			
Female				
Male				
Right vs. Left	0.438			
Right-sided				
Left-sided				
Postoperative complications	*0.024*			** *0.024* **
No		1		
Yes		0.587	0.370–0.932	
Major complications	0.441			
No				
Yes				
Extrahepatic metastases	*0.003*			** *0.015* **
No		1		
Yes		0.407	0.197–0.839	
Extension of hepatectomy	0.481			
Minor				
Major				
Associated ablation	0.918			
No				
Yes				
Maximum size of CLMs	0.636			
≤5 cm				
>5 cm				
Maximum size of CLMs	0.87			
≤3 cm				
>3 cm				
Number of CLMS	*0.026*			0.057
Single		1		
Multiple		0.627	0.387–1.015	
Number of CLMS	0.971			
<4				
≥4				
TBS	0.472			
≤4.47				
>4.47				
CLMs’ distribution	0.126			
Unilobar				
Bilobar				
T-status	0.903			
T1–T3				
T4				
N-status	0.188			
N−				
N+				
Synchronous vs. Metachronous	0.315			
Synchronous				
Metachronous				
Initially resectable CLMs	0.716			
Yes				
No				
Preoperative chemotherapy	0.123			
No				
Yes				
Adjuvant chemotherapy	0.383			
Yes				
No				

Italic-bold: for all values lower than 0.05 (in this column).

**Table 4 medicina-58-01100-t004:** Univariate and multivariate analysis for SAR in RAS-wt patients.

Clinico-Pathologic Characteristics	*p* Value (Univariate, Log-Rank)	HR (Multivariate, Cox Regression)	95% CI (Multivariate, Cox Regression)	*p* Value (Multivariate, Cox Regression)
Age	0.248			
≤65 y-o				
>65 y-o				
Sex	0.796			
Female				
Male				
Right vs. Left	*<0.001*			** *<0.001* **
Right-sided		0.222	0.102–0.483	
Left-sided		1		
Postoperative complications	0.415			
No				
Yes				
Major complications	0.343			
No				
Yes				
Extrahepatic metastases	0.237			
No				
Yes				
Extension of hepatectomy	0.187			
Minor				
Major				
Associated ablation	0.161			
No				
Yes				
Maximum size of CLMs	0.738			
≤5 cm				
>5 cm				
Maximum size of CLMs	0.419			
≤3 cm				
>3 cm				
Number of CLMS	0.784			
Single				
Multiple				
Number of CLMS	0.464			
<4				
≥4				
TBS	0.389			
≤4.47				
>4.47				
CLMs’ distribution	0.496			
Unilobar				
Bilobar				
T-status	0.321			
T1-T3				
T4				
N-status	*0.011*			** *0.007* **
N−				
N+		0.407	0.211–0.786	
Synchronous vs. Metachronous	0.447			
Synchronous				
Metachronous				
Initially resectable CLMs	0.192			
Yes				
No				
Preoperative chemotherapy	0.746			
No				
Yes				
Adjuvant chemotherapy	0.285			
Yes				
No				
Resection of recurrence	*0.007*			0.123
No		0.621	0.339–1.138	
Yes		1		
Time to recurrence	*0.048*			0.25
≤12 months		0.677	0.349–1.315	
>12 months		1		

Italic-bold: for all values lower than 0.05 (in this column).

## Data Availability

The datasets are not publicly available but are available from the corresponding author on reasonable request.

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
