# Peer review of "Embryologic Origin of the Primary Tumor and RAS Status Predict Survival after Resection of Colorectal Liver Metastases"

_medicina, 2022, doi:10.3390/medicina58081100_

Round 1

Reviewer 1 Report

Dr Ioana Mihaela Dinu and colleagues report on their study focused on embryologic origin of the primary tumor found to be an independent prognostic factor for overall survival in patients with RAS-wt resected colorectal liver metastases. The manuscript is dedicated to the important medical problem of patients’ survival in devastative cancerous disease. No major concerns may be highlighted. Minor modifications are to be made. Firstly, the title is overloaded by words representing a phrase, which is grammatically poorly composed. It is to rephrase the title; it should be more concise and direct. Secondly, Methods part requires an addition; molecular and cellular aspects of the study should be made as a separate sub-part of the Methods and described in full details. Conclusions should be redesigned to become more concise and direct or, in other words, more understandable. One may recommend making point-by-point conclusions (e.g. numbered conclusions).

Reviewer 2 Report

The manuscript entitled “Embryologic origin of the primary tumor is an independent prognostic factor for overall survival and survival after recurrence only in patients with RAS-wt resected colorectal liver metastases” is interesting and can be accepted for publication after these changes.

The title of manuscript is not attractive. The title seems to be explaining the results. The “overall survival and survival after recurrence” also do not seem suitable here. Authors can use such words “To predict survival etc…

In introduction, the authors should start with the explanation of introduction and epidemiology of colorectal cancer.

In Materials and Methods, could you please explain how the patients were diagnosed for colorectal liver metastases. Histology? Radiology?

The treatment allocation is not clear. Whether all the patients undergone similar set of treatment? The data described here ranges from 2006 to 2019. Does the treatments were same during the 14 years? The treatment modalities might be affecting the overall survival (OS) or relapse-free survival (RFS) rather than the RAS status. Please clarify

Round 2

Reviewer 2 Report

The authors have improved the paper and can be accepted for publication.